# Bright high-order harmonic generation with controllable polarization from a relativistic plasma mirror

Zi-Yu Chen[1,2] & Alexander Pukhov[1]

Ultrafast extreme ultraviolet (XUV) sources with a controllable polarization state are powerful tools for investigating the structural and electronic as well as the magnetic properties of materials. However, such light sources are still limited to only a few free-electron laser facilities and, very recently, to high-order harmonic generation from noble gases. Here we propose and numerically demonstrate a laser–plasma scheme to generate bright XUV pulses with fully controlled polarization. In this scheme, an elliptically polarized laser pulse is obliquely incident on a plasma surface, and the reflected radiation contains pulse trains and isolated circularly or highly elliptically polarized attosecond XUV pulses. The harmonic polarization state is fully controlled by the laser–plasma parameters. The mechanism can be explained within the relativistically oscillating mirror model. This scheme opens a practical and promising route to generate bright attosecond XUV pulses with desirable ellipticities in a straightforward and efficient way for a number of applications.

[1] Institut für Theoretische Physik I, Heinrich-Heine-Universität Düsseldorf, Düsseldorf D-40225, Germany. [2] National Key Laboratory of Shock Wave and Detonation Physics, Institute of Fluid Physics, China Academy of Engineering Physics, Mianyang 621999, China. Correspondence and requests for materials should be addressed to Z.-Y.C. (email: ziyu.chen@uni-duesseldorf.de) or to A.P. (email: pukhov@tp1.uni-duesseldorf.de).

Ultrafast radiation sources in the extreme ultraviolet (XUV) range have become a major tool to study electronic structures and the dynamics of atoms, molecules and condensed matter. Because polarization is a fundamental property of light and controls its interaction with matter, it is particularly important that these light sources have a tunable polarization. Furthermore, polarization control opens a wider range of applications. For instance, the magnetic as well as the electronic and phononic properties of materials can be studied using circularly polarized (CP) or elliptically polarized (EP) XUV pulses using techniques such as magnetic circular dichroism spectroscopy[1]. The magnetic circular dichroism technique has proven to be very useful to probe the spin-resolved features in magnetic materials in an element-specific manner[2], and thus is of great interest for understanding correlated systems in condensed matter physics. In addition, CP/EP XUV pulses also enable a unique probe of chiral molecules[3,4], for example, measuring the photoionization process via photo-electron circular dichroism[5]. As such, CP/EP XUV pulses also find a wide range of applications in studying chemical and biological systems.

To date, significant efforts have been devoted to generating the ultrafast XUV with a variable polarization state. The first free-electron laser facility that was specially designed to produce such a light source, named Free Electron laser Radiation for Multidisciplinary Investigations (FERMI, Trieste, Italy), has become accessible very recently[6]. Polarization control at FERMI is achieved by adjusting the configuration of the undulators. Although powerful, these large-scale facilities are expensive and complex, thus limiting their wide accessibility. Therefore, there remains a strong need for sources of coherent CP/EP XUV radiation at the table-top scale.

High-order harmonic generation (HHG) from noble gases has been explored extensively as a route to generate an ultrafast XUV source[7]. This mechanism, however, encounters intrinsic difficulties in generating a CP XUV pulse. This is because the HHG is based on the tunnel ionization, acceleration and recombination of electrons ripped from an atom in the presence of a laser field, which is explained by the so-called three-step model[8]. As a consequence, the emission of HHG decreases exponentially with increasing the laser ellipticity because the lateral motion of the detached electron induced by the ellipticity makes the electron less likely to recollide with its parent ion. To be exact, the electron never returns to the parent ion with a CP driving laser. To overcome this drawback, several techniques have been proposed and demonstrated recently to generate quasi-CP or highly EP HHG, such as using pre-aligned molecule targets[9], resonant HHG in EP laser fields[5], a co-propagating bi-chromatic EP or CP driving laser with opposite helicity[1,10–12] and a co-propagating bi-chromatic linearly polarized (LP) driving laser with orthogonal polarization[13]. However, owing to their low ionization thresholds and conversion efficiencies, these sources typically suffer from low photon yields.

To fill the gap between large-scale facilities and HHG from gas, XUV via HHG[14] and other mechanisms[15–18] from laser-irradiated plasma surfaces offers a promising alternative to generate an XUV source with high brightness. In principle, with plasma targets there is no limitation on the applicable laser intensity and thus the XUV intensity[14]. Several radiation mechanisms have been theoretically and experimentally identified as responsible for the HHG process, including coherent wake emission (dominant in the weakly relativistic regime)[19–22], relativistically oscillating mirror (ROM; dominant in the strongly relativistic regime)[23–28] and coherent synchrotron emission[29–31]. It has been demonstrated that LP HHG can be generated relatively efficiently using an LP driving laser at oblique incidence.

It has been commonly assumed up to now that the ROM mechanism fails for a CP driving laser. Moreover, a polarization gating (aka relativistic coherent control) has been proposed to select a single LP attosecond pulse from a pulse train[32–34]. According to the ROM theory[25], these attosecond pulses are emitted when electrons at the plasma surface are moving towards the observer and their tangential momenta vanish. This is never the case for an EP laser pulse normally incident on the plasma surface, which causes HHG to be strongly suppressed. For an EP laser pulse at oblique incidence, two experimental groups have observed HHG at relatively small angles of incidence[35,36]. However, only the harmonic intensities were measured, with no information about the harmonic polarization states.

In this paper, we propose and numerically demonstrate the generation of intense HHG with fully controlled polarization from laser plasmas. We show that this can be achieved using a CP laser obliquely incident onto a plasma surface. Both pulse trains and isolated circular or highly elliptic attosecond XUV pulses can be obtained. By changing the incidence angle, the harmonic polarization state can be tuned from quasi-circular through elliptical and linear to an elliptical polarization of opposite helicity. Switching the helicity of the incident laser, the handedness of the harmonics can be easily reversed. The scheme works for a wide range of laser and plasma parameters, and the efficiency is comparable to that using an LP laser. This very promising new procedure thus provides a straightforward and efficient way to obtain a bright attosecond XUV source with desirable ellipticities and holds the potential of making a very large avenue of research more accessible for a number of laser laboratories worldwide.

## Results

**Scheme**. Figure 1a shows the scheme of the proposed configuration for the HHG with a desired polarization state. The basic idea is to use a CP relativistic laser pulse obliquely incident on a solid–plasma surface using a radiation mechanism known as the ROM model. In the ROM model, under the combined action of the ponderomotive force of the laser and the electrostatic restoring force resulting from charge separation, the surface electrons oscillate with relativistic speeds and reflect the laser pulse like mirrors. During this nonlinear process, harmonics of the fundamental laser frequency are generated as a result of Doppler up-shifting. Except for some special cases (that is, few-cycle laser pulse interactions with near-critical density plasmas[37]), normal incidence CP laser pulses cannot generate harmonics for two reasons. Firstly, CP laser pulses lack the fast oscillating component in the ponderomotive force. Secondly, driven by CP pulses, electrons always have a relativistically large tangential momentum: when one tangential component vanishes, the other reaches its maximum. As a result, electrons never move towards the observer and do not emit high harmonics efficiently. This difference between CP and LP pulses forms the basis of the polarization gating[32–34], which is the method proposed herein to obtain an isolated single attosecond pulse from a train of attosecond pulses of ROM harmonics. This is true for a CP laser pulse at normal incidence.

However, for oblique incidence interactions, HHG can be efficiently generated even by CP laser pulses. The force acting on the plasma surface does contain a fast oscillating component owing to the normal (p-polarization) component of the laser electric field. Further, the oblique incidence can be reduced to a normal incidence case using Lorentz transformation to a moving frame of reference where plasma is streaming along the surface (see Methods section). In this frame of reference, electrons have an initial tangential momentum. When the angle of incidence is

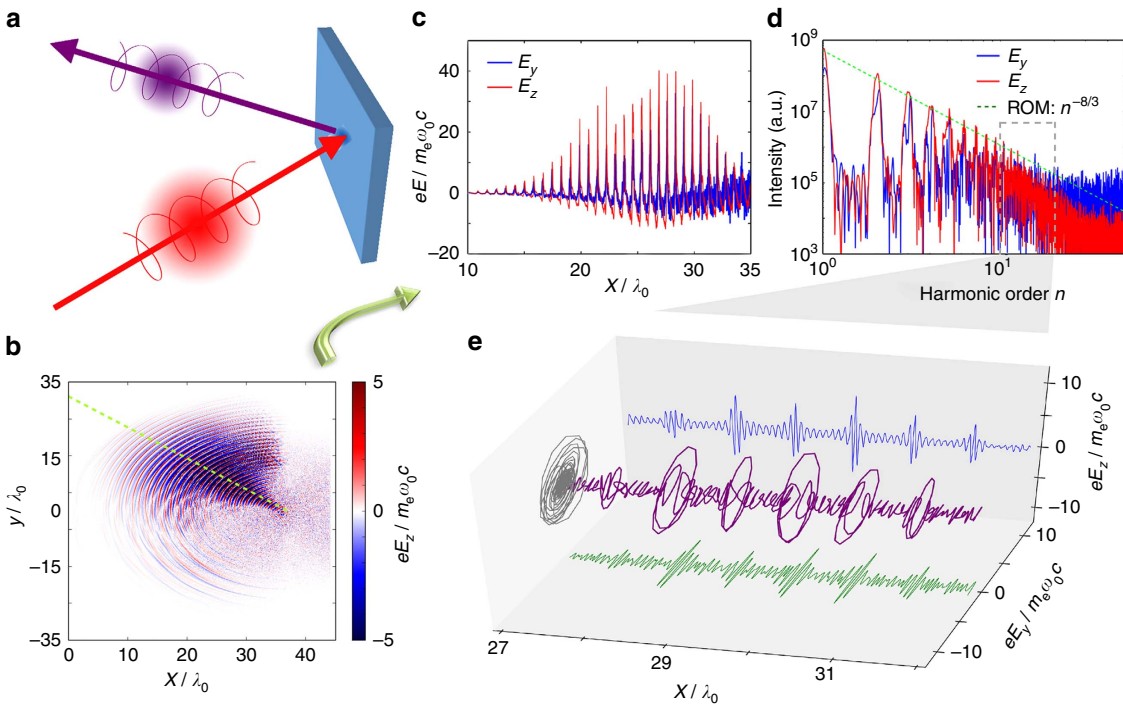

**Figure 1 | Scheme and 2D simulation results.** (**a**) The proposed experimental configuration for generation of the polarization-controlled harmonics by a CP laser pulse obliquely incident on a plasma surface. The red and purple arrows represent the incident laser pulse and the reflected XUV harmonics, respectively. (**b**) A snapshot of the electric field component $E_z$ of the reflected pulse from the 2D simulation results at time $t = 36T_0$. The green dashed line marks the specular reflection direction. (**c**) Temporal waveform and (**d**) the corresponding Fourier spectra of the reflected pulse along the specular reflection. The green dashed line in panel (**d**) corresponds to the predicted scaling law $I_{ROM}(n) \propto n^{-8/3}$ by the ROM theory. (**e**) The reconstructed 3D image of the electric field vector of the attosecond pulses (purple), obtained after spectral filtering by selecting the 10th–20th harmonic orders (indicated by the harmonics within the dashed grey box in panel **d**). Waveform of the two orthogonal electric field components $E_y$ (green) and $E_z$ (blue), as well as the projection of $E_y - E_z$ (grey), are also shown.

adjusted correctly, this momentum can exactly compensate for the momentum induced by the laser field so that there exist moments when the plasma surface electrons move exactly towards the observer and reflect the CP laser. However, the different laser polarizations may have different phase lags at the nonlinear reflection from the plasma, which makes it necessary to perform simulations to clarify whether the properties of the incident laser, such as polarization and coherence, are preserved.

**Two-dimensional simulation results**. We first carried out two-dimensional (2D) particle-in-cell simulations to show a general picture of the ellipticity of HHG from a CP laser obliquely irradiated onto plasma surfaces. The laser and plasma parameters are chosen to match realistic experiments, wherein the laser with a normalized amplitude of $a_0 = 30$ is obliquely incident at an angle of $\theta = 40°$ onto a plasma of density $n_e = 100n_c$, where $n_c$ is the critical density (see Methods section). Figure 1b presents a snapshot of the electric field component $E_z$ of the reflected pulse in the $x - y$ plane at time $t = 36T_0$, where $T_0$ is the laser period. The green dashed line marks the direction of specular reflection of the incident laser. A temporal waveform of the radiation in the specular reflection is shown in Fig. 1c, where both the $E_y$ and $E_z$ components are depicted. It is apparent that both $E_y$ and $E_z$ have an amplitude level the same as that of the incident laser.

Figure 1d shows the Fourier spectra corresponding to Fig. 1c. The green dashed line corresponds to the scaling law for the spectral intensity $I_{ROM}$ as a function of the harmonic order $n$: $I_{ROM}(n) \propto n^{-8/3}$, which is given by the Baeva–Gordienko–Pukhov (BGP) theory[25] of ROM. The excellent agreement of

the spectra with the theoretically predicted power law suggests that the HHG mechanism here is within the ROM regime. Harmonic structures up to the 20th order can be clearly observed for both the $E_y$ and $E_z$ components. Beyond that, however, the spectral line structure is not periodic, indicating that the periodicity of the attosecond pulses changes with time. It is worth noting that the 2D simulations for HHG from solid–plasma surfaces are computationally expensive and the resolution is limited. Here, we only resolve the HHG up to the 20th order for demonstration purposes, but harmonic spectra with well-defined periodic structures up to much higher orders can be generated and have been observed experimentally. For example, well-defined harmonic structures up to at least the 46th order have been observed with almost the same laser (excepting that the polarization is linear) and plasma parameters[38]. Another experiment with a much lower intensity of $a_0 = 3.5$ has also demonstrated that harmonic comb structures up to about the 40th order can be observed[39]. Therefore, periodic harmonic orders higher than the 20th can be expected. In addition, different harmonic orders are appropriate for different applications. For example, the harmonics of the 7th–20th orders (photon energies around 10–30 eV) are of particular interest for studies such as molecular photoionization, because this frequency range is close to the ionization thresholds of most molecular systems[5]. Also, harmonics of the 35th–42nd orders (photon energies around 55–65 eV) are required for investigating the magnetic properties of solids, because this frequency range covers the $M$ absorption edges of the magnetic elements Fe, Co and Ni (ref. 40). We leave the HHG with higher orders to be investigated with one-dimensional (1D) simulations later.

Applying a band-pass spectral filter that selects harmonics between the 10th and 20th orders, we obtain a train of attosecond XUV pulses, as shown in Fig. 1e. From the helical structures of the electric field contour $E^H = E_y^H + E_z^H$ plotted in this three-dimensional (3D) image, we can see directly that each attosecond HHG pulse is elliptically polarized. The HHG pulses reach a peak electric field amplitude of $E^H = 5$ (normalized to $m_e \omega_0 c/e \approx 4 \times 10^{12}$ V m$^{-1}$), which corresponds to a dimensional value of $E^H = 2 \times 10^{13}$ V m$^{-1}$. This clearly demonstrates the potential of the ROM mechanism to obtain a bright helical XUV source. The averaged amplitude ratio between the two electric components in this frequency range is $\varepsilon = \min\left\{E_y^H, E_z^H\right\} / \max\left\{E_y^H, E_z^H\right\} = 0.96$, indicating that a high ellipticity can be reached. The phase shift $\Delta\phi^H$ between the two electric field components $\phi_z^H$ and $\phi_y^H$ is $\Delta\phi^H = \phi_z^H - \phi_y^H = 0.36\pi$ for the HHG pulse around $x = 30\lambda_0$ in Fig. 1e. The sign of $\Delta\phi^H$ also demonstrates that the HHG pulse generated here has the same helicity as the incident laser pulses $\left(\phi_z^L - \phi_y^L = \pi/2\right)$.

**Parametric study.** In the following, we use a series of 1D simulations with a higher resolution to study the parametric dependence of the HHG ellipticity. From Fig. 2a we can see that, for a broad range of laser amplitudes from $a_0 = 5$ to $a_0 = 30$, both the amplitude ratio $\varepsilon$ and the phase shift $\Delta\phi^H$ between the two electric components of HHG change insignificantly. Similarly, $\varepsilon$ and $\Delta\phi^H$ exhibit only a weak dependence on the initial plasma density in the range of 50–400$n_c$ (see Fig. 2b). Compared with the laser amplitude and plasma density, the plasma density scale length $L_s$ plays a more important role in changing the HHG amplitude ratio $\varepsilon$, as shown in Fig. 2c. For laser plasma interactions in the ultrarelativistic regime ($a_0 \gg 1$), the dynamics are largely determined by the dimensionless similarity parameter

$S = n_e/a_0 n_c$. Because the reflection mainly occurs at the critical surface with a relativistic density of $n_c^{Rel} \approx n_c a_0$, that is, at a fixed $S = 1$, the system is expected to be self-similar[41]. Therefore the dynamics of harmonic generation do not depend separately upon $a_0$ and $n_e$, but instead upon the scale length through the electron density profile $n_e = n_c^{Rel} \exp^{-\sqrt{Sx}/L_s}$ (ref. 38). A previous study has suggested that there exists an optical scale length whose value is about $c/\omega_0$ (ref. 38), where $c$ is the light speed in vacuum and $\omega_0$ is the laser angular frequency. The parametric study here shows that the helical HHG exists for a wide range of laser and plasma parameters given that the scale length is well controlled. Furthermore, the feasibility of scale length control is confirmed in experiments[38].

**Polarization control.** Based on the parametric studies above, we choose an optimal scale length of $L_s = 0.1$, a moderate laser amplitude of $a_0 = 5$ and a plasma density of $n_e = 200n_c$ to study polarization state controllability in the HHG pulse. Figure 2d shows the amplitude ratio and phase shift of the HHG field components as a function of the laser incidence angle $\theta$. The HHG in the frequency range of the 20th–30th orders are selected, except that the 5th–10th orders for $\theta = 22.5°$ and the 15th–20th orders for $\theta = 40°$ are used owing to a lower cutoff of well-defined harmonic structures at these relatively small incidence angles. Nevertheless, we found that, even when using the higher orders of 35th–42th in the case of $\theta = 40°$, elliptical HHG pulses can be generated, although with a smaller ellipticity value $\varepsilon$. Notably, an amplitude ratio of $\varepsilon$ close to unity, together with a phase shift of $\phi^H \approx \pi/2$, is obtained at the angle of $\theta = 22.5°$. This indicates that intense quasi-CP HHG pulses are generated with this simple geometry. Moreover, as the angle of incidence $\theta$ increases, the phase shift $\Delta\phi^H$ changes continuously from $+\pi/2$ to 0, and eventually to $<-\pi/2$. This signifies that the polarization state of the HHG pulses varies as $\theta$ increases, from a polarization state

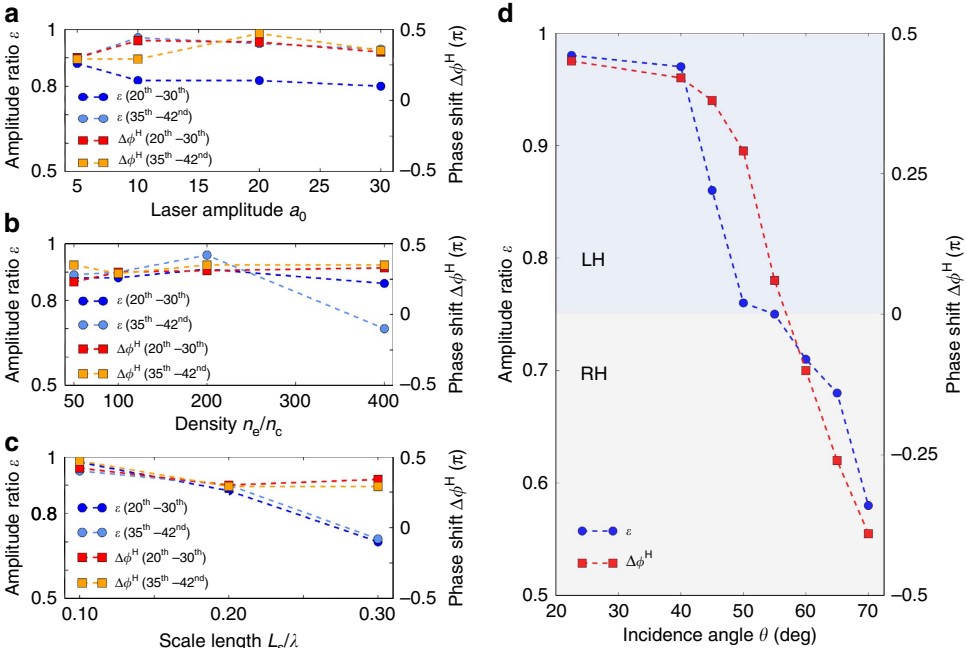

**Figure 2 | Parametric study and polarization control. (a–d)** Amplitude ratio $\varepsilon$ and phase shift $\Delta\phi^H$ between the two orthogonal components of the harmonic electric fields as a function of (**a**) laser amplitude $a_0$, (**b**) plasma density $n_e$, (**c**) plasma density scale length $L_s$ and (**d**) laser incidence angle $\theta$. The other parameters are: (**a**) $n_e = 100n_c$, $\theta = 40°$ and $L_s = 0.2$; (**b**) $a_0 = 5$, $\theta = 40°$ and $L_s = 0.2$; (**c**) $a_0 = 5$, $\theta = 40°$ and $n_e = 100n_c$; and (**d**) $a_0 = 5$, $n_e = 200n_c$ and $L_s = 0.1$. The $L_s$ value is normalized by $\lambda$ in the moving frame for convenience in the 1D simulations. The different shaded areas in panel (**d**) represent harmonics possessing opposite helicities. LH, left-handed; RH, right-handed.

that is circular ($\Delta\phi^H = \pi/2$) through one that is elliptical ($0 < \Delta\phi^H < \pi/2$) and linear ($\Delta\phi^H = 0$) and finally to one that is elliptical of opposite helicity ($-\pi/2 < \Delta\phi^H < 0$). In this way, therefore, a practical and straightforward method to control the ellipticity of the HHG pulses is produced by simply adjusting the incidence angle of the laser pulse, which is important to a number of applications.

**Circular attosecond pulses using elliptic laser**. In addition to obtaining quasi-circular or highly elliptic HHG using CP laser pulses at a small-angle incidence, here we show CP harmonics and/or attosecond XUV pulses can also be generated using EP laser pulses at an oblique incidence. Considering the above case of $\theta = 40°$ in Fig. 2d for example, the HHG amplitude ratio is $\varepsilon = 0.97$ and the phase shift is $\Delta\phi^H = 0.42\pi$. For a perfect CP attosecond XUV pulse; however, the phase shift must be $\Delta\phi^H = \pi/2$; we therefore compensate for the necessary additional phase shift of $\delta\phi = 0.08\pi$ by using EP laser pulses that have a phase shift of $\phi_z^L - \phi_y^L = \pi/2 + \delta\phi = 1.822$. The other parameters are the same as those used in the case of $\theta = 40°$ in Fig. 2d. The waveform of the generated attosecond XUV pulse train is given in Fig. 3a, showing a pulse train whose amplitude ratio is $\varepsilon \cong 1.0$ and the phase shift is $\Delta\phi^H \cong \pi/2$. As a result, an attosecond XUV pulse train with nearly perfect circular polarization has been generated using EP laser pulses. This approach is also very promising and is easy to implement experimentally.

**Isolated attosecond helical XUV pulse**. Attosecond HHG pulse trains have proven to be useful in studying ultrafast XUV nonlinear processes, such as the photodissociation of molecules[42] and chiral experiments[1]. However, an isolated single attosecond helical XUV pulse holds the potential for time-resolved dichroism measurements with unprecedented temporal resolutions[43]. For instance, important questions regarding the timescale of magnetization dynamics in correlated materials may be addressed with such a tool[40]. Here, we demonstrate how an isolated single attosecond elliptical/circular HHG pulse can be generated with the present scheme using a few-cycle laser pulse. Figure 3b shows a resulting waveform of the attosecond HHG pulse after spectral filtering whereby the 35th–42nd orders are selected. Here, the incident EP laser pulse has a duration of 5 fs full-width at half-maximum, an amplitude of $a_0 = 5$, an initial phase of $\Delta\phi^L = 1.822$ and an incidence angle of $\theta = 40°$, while the plasma density is $n_e = 200 n_c$. It can be seen that an isolated single attosecond XUV pulse with quasi-circular polarization has been

generated. This shows the possibility of applying this technique to ultrafast dichroism measurements.

**Switching HHG handedness**. For dichroism study applications, the difference in the absorption of left-handed (LH) and right-handed (RH) light is measured; that is, $\Delta A(\lambda_L) = A_{LH}(\lambda_L) - A_{RH}(\lambda_L)$, where $\lambda_L$ is the light wavelength. Thus it is important to generate helical light with opposite handedness. This can be easily achieved by changing the handedness of the incident laser pulse in our scheme because the Vlasov–Maxwell equations are symmetric about the handedness of electromagnetic fields. To demonstrate this, we compared the HHG produced by CP laser pulses possessing opposite handedness, where both laser pulses possess an amplitude of $a_0 = 5$ and are obliquely incident at the angle of $\theta = 40°$ onto a plasma of density $n_e = 200 n_c$. Figures 4a,b show the HHG waveform generated by an LH laser and an RH laser, respectively, where a band-pass filter was used to select the 15th–20th harmonic orders. It is seen that the two HHG pulses are nearly the same except for the opposite phase shift $\Delta\phi^H$. This shows the feasibility of reversing the rotation direction of the HHG pulse by simply switching the handedness of the incident CP laser pulses.

**HHG efficiency**. As is known, the HHG efficiency using a CP laser is low at normal or small-angle incidence[32–34]. This result changes drastically, however, as the angle of incidence increases. Experimental results by Yeung et al. have shown that at 22.5° incidence the harmonic (13th–28th) efficiency using a CP laser is at least two orders of magnitude lower than that using an LP laser[36]. On the other hand, experimental results by Easter et al. have shown that at 35° incidence the harmonic (13th–19th) efficiency using a CP laser is just a factor of 3 lower than that using an LP laser with the same harmonic orders[35]. To verify the dependence of the efficiency upon the incidence angle, we carry out a series of simulations, as shown in Fig. 5a, in which the laser amplitude is $a_0 = 5$, the plasma density is $n_e = 200 n_c$ and the harmonic orders are 13th–30th. At incidence angles of 22.5° and 35°, the simulation results are in excellent agreement with the above-mentioned experimental results[35,36]. In addition, the simulation results predict that the harmonic efficiency with the CP laser increases with the angle of incidence, and reaches the same value as when using an LP laser at 45° incidence. As the incidence angle is further increased, the efficiencies with the CP and the LP lasers tend to stay at the same level. Figure 5b shows the $E_y$ components of the HHG spectra when using CP and LP lasers at $\theta = 55°$ incidence, where it is also seen that the HHG efficiency

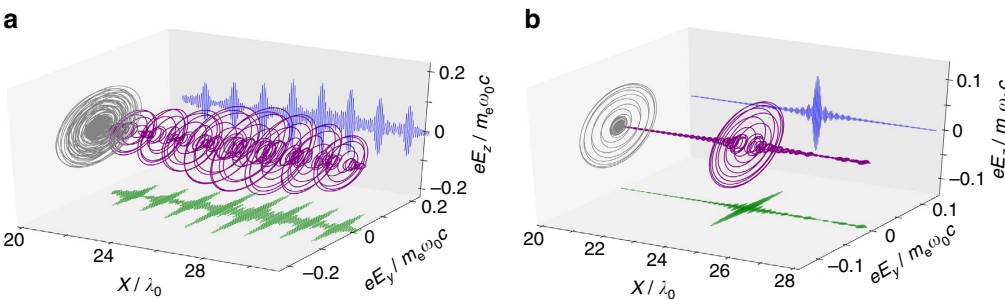

**Figure 3 | Attosecond helical XUV pulses.** (**a**) Waveform of a CP XUV attosecond pulse train after spectral filtering where the 15th–20th harmonic orders are selected. Here an EP laser pulse of 30 fs duration is used. (**b**) Waveform of a CP XUV isolated single attosecond pulse after spectral filtering where the 35th–42nd harmonic orders are selected. Here a few-cycle EP laser pulse of 5 fs duration is used. The other parameters are: laser amplitude $a_0 = 5$; laser initial phase $\phi_z^L - \phi_y^L = 1.822$; laser incidence angle $\theta = 40°$; plasma density $n_e = 200 n_c$; and plasma scale length $L_s = 0.1$. In both panels, waveform of the 3D electric field vector (purple), the two orthogonal electric field components $E_y$ (green) and $E_z$ (blue) and the projection of $E_y - E_z$ (grey) are displayed.

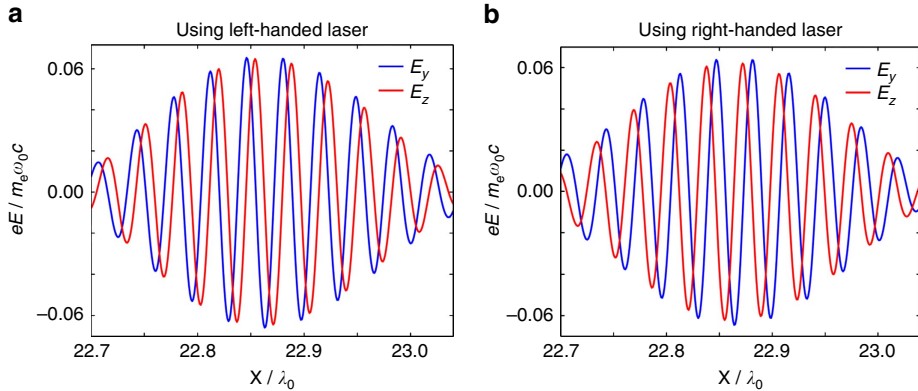

**Figure 4 | Switching the harmonic handedness.** Electric field waveform of the harmonics using a laser with helicity of (**a**) left-handedness and (**b**) right-handedness. Spectral filtering is applied where the 15th–20th harmonic orders are selected. The laser amplitude is $a_0 = 5$ and the incidence angle is $\theta = 40°$. The plasma density is $n_e = 200 n_c$ with a scale length of $L_s = 0.1$.

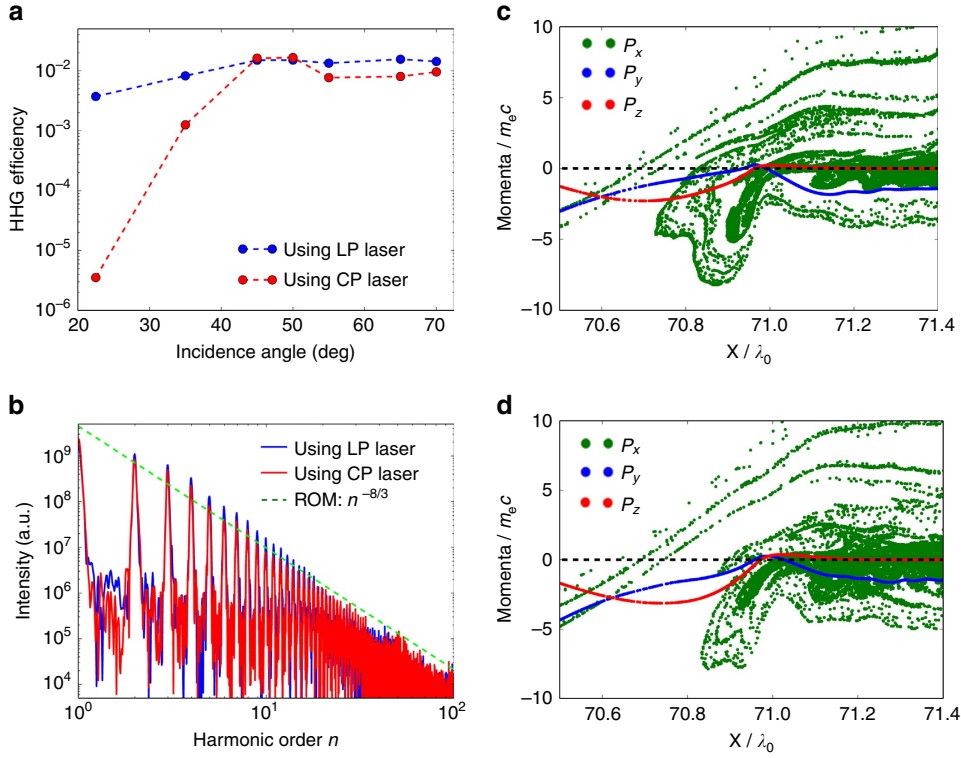

**Figure 5 | Harmonic efficiency and plasma dynamics.** (**a**) Influence of the incidence angle upon the efficiency of the harmonics (13th–30th) with CP and LP laser pulses. The laser amplitude is $a_0 = 5$ for the CP laser and $a_0 = 7.07$ for the LP laser to keep the intensity and pulse energy the same. (**b**) HHG spectra of $E_y$ component, compared between the cases using CP and LP laser pulses at 55° incidence. The green dashed line corresponds to the predicted scaling law $I_{\mathrm{ROM}}(n) \propto n^{-8/3}$ by the ROM theory. (**c,d**) Spatial distribution of electron longitudinal ($p_x$) and transverse momenta ($p_y$ and $p_z$) for the case of a CP laser at 55° incidence at two different times of (**c**) $t = 28.64 T_0$ and (**d**) $t = 30.40 T_0$. In these simulations, the other parameters are: laser amplitude $a_0 = 5$, plasma density $n_e = 200 n_c$ and scale length $L_s = 0.1$. The momenta are normalized by $m_e c$. The dashed black lines in panels (**c,d**) mark the zero momenta.

with the CP laser is comparable to that with the LP laser. These results show the potential of achieving efficient helical HHG with the present scheme. Moreover, as mentioned, intense HHG can be generated even with a low efficiency, because the applied laser intensity is high.

**Regime of validity.** Figure 5b again shows that the harmonic spectra agree well with the $I_{\mathrm{ROM}}(n) \propto n^{-8/3}$ scaling law, which is

predicted by the BGP theory (also termed the $\gamma$-spikes model) of ROM[25]. This theory implies the existence of zeros in the transverse momenta of plasma surface electrons[44]. To further demonstrate that the HHG mechanism here is within the $\gamma$-spikes ROM regime, we plot the spatial distribution of the electron longitudinal ($p_x$) and transverse momenta ($p_y$ and $p_z$) at two different times from the simulation results with the CP laser at 55° incidence, as shown in Fig. 5c,d. It can be clearly seen that moments do exist when both of the transverse momenta of the

plasma surface electrons become zero simultaneously, and it is at these moments that the harmonics are efficiently emitted. In addition, the parametric studies above also show that our scheme works well for a broad range of laser amplitudes ($a_0$ from 5 to 30) at oblique incidence. These results suggest that the mechanism here is in accordance with the $\gamma$-spikes model of ROM, and thus is relevant to the strongly relativistic regime with $a_0^2 \gg 1$ and relativistically overdense plasmas with $S \gg 1$.

In summary, a scheme to generate CP or highly EP attosecond XUV pulses is proposed and numerically demonstrated, which is based on high harmonic generation from a relativistic plasma mirror. It is shown that such harmonics can be efficiently generated when the laser–plasma parameters are suitable. In addition, the harmonic polarization is fully controllable by the laser–plasma parameters. The scheme allows the use of a relativistically intense laser, and thus it is a promising scheme to achieve a chiral XUV source with high brilliance. This provides an exciting tool with applications in a number of fields.

## Methods

**Particle-in-cell simulation.** We carried out all simulations using the Virtual Laser Plasma Lab (VLPL) code[45]. For 2D simulations, the size of the simulation box is $45\lambda_0 \times 70\lambda_0$ in the $x - y$ plane, with a laser wavelength of $\lambda_0 = 800$ nm and a cell size of $\lambda_0/200$ in each dimension. The laser and plasma parameters are chosen to match those used in the experiments[38]. The laser pulse has a normalized amplitude of $a_0 = eE_0/m_e\omega_0 c = 30$ (corresponding to an intensity of $2 \times 10^{21}$ W cm$^{-2}$) and pulse duration of 30 fs full-width at half-maximum, where $E_0$ is the laser electric field amplitude, $e$ is the elementary charge and $m_e$ is the electron mass. The pulse is focused into a Gaussian spot with a diameter of 2 μm, which requires a Ti:sapphire laser system that can deliver a pulse energy of about 1 J. The laser pulse is obliquely incident at an angle of $\theta = 40°$ onto the target, which is taken to be a fully ionized plasma. The plasma slab has an electron density of $n_e = 100n_c$ and a thickness of 500 nm, where $n_c = m_e\omega_0^2/4\pi e^2$. In the front of the plasma slab, preplasma exists with an exponential density profile and a density scale length of $L_s = 0.2\lambda_0$. To simulate oblique laser incidence in the 1D setup, a Lorentz transformation from the laboratory frame to a moving frame of reference has been made[24,46]. As such, the laser is transformed to be at normal incidence onto a plasma slab streaming in the $y$ direction parallel to the planar surface. For all 1D simulations, a relatively high spatial resolution of 1,000 cells per laser wavelength in the moving frame is used.

**Control of polarization.** A CP laser pulse can be represented as a superposition of two LP pulses with equal amplitude and a constant phase difference of $\pi/2$: $\mathbf{E}_{CP} = \mathbf{E}_y + \mathbf{E}_z$ with $\mathbf{E}_y = E_0 \cos(\omega_0 t)\hat{\mathbf{e}}_y$ and $\mathbf{E}_z = E_0 \sin(\omega_0 t)\hat{\mathbf{e}}_z$, where $\hat{\mathbf{e}}_y$ and $\hat{\mathbf{e}}_z$ are respectively the unit vector along the $y$ and $z$ directions. Thus the corresponding vector potential can be written as $\mathbf{A}_y = A_0 \sin(\omega_0 t)\hat{\mathbf{e}}_y$ and $\mathbf{A}_z = -A_0 \cos(\omega_0 t)\hat{\mathbf{e}}_z$. In the 1D geometry, the canonical momentum in the transverse direction is conserved: $\mathbf{p}_\perp - e\mathbf{A}_\perp/c = $ constant, where $\mathbf{p}_\perp$ and $\mathbf{A}_\perp$ are the transverse momentum and vector potential, respectively. In the moving frame of reference, the initial momenta in the transverse directions are $\mathbf{p}_{y0} = -m_e c \tan\theta\hat{\mathbf{e}}_y$ and $\mathbf{p}_{z0} = 0$. Then we can obtain the expression for the transverse momentum as:

$$\mathbf{p}_y = -m_e c \tan\theta\hat{\mathbf{e}}_y + e\mathbf{A}_y/c = (-m_e c \tan\theta + eA_0 \sin(\omega_0 t)/c)\hat{\mathbf{e}}_y \quad (1)$$

$$\mathbf{p}_z = e\mathbf{A}_z/c = -eA_0 \cos(\omega_0 t)/c\hat{\mathbf{e}}_z \quad (2)$$

According to the BGP theory[25], the HHG is emitted when the transverse momentum of the surface electron $\mathbf{p}_\perp$ reaches a minimum or vanishes. In the case of a laser normally incident with $\theta = 0$, we observe that $\mathbf{p}_\perp = eA_0/c(\sin(\omega_0 t)\hat{\mathbf{e}}_y - \cos(\omega_0 t)\hat{\mathbf{e}}_z)$, which never vanishes or reaches a minimum, and consequently, no harmonics are generated. The situation changes with an oblique angle of incidence $\theta$, which makes it possible to have $\mathbf{p}_y$ and $\mathbf{p}_z$ simultaneously reach a minimum or vanish. The incident angle $\theta$ therefore provides a degree of freedom that can be used to adjust the relative amplitude and phase between the two components of the transverse momentum, and thus can be used to change the polarization state of the harmonics generated.

**Data availability.** The data that support the findings of this study are available from the corresponding authors upon request.

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

## Acknowledgements

Z.Y.C. acknowledges financial support from the China Scholarship Council (201404890001). This work was supported by the Deutsche Forschungsgemeinschaft SFB TR 18, EU FP7 project EUCARD-2 and the Science and Technology Fund of the National Key Laboratory of Shock Wave and Detonation Physics (China) with project Nos. 077110 and 077160.

## Author contributions

Z.Y.C. conceived and conducted the simulations, and drafted the manuscript. A.P. developed the code and theory, and supervised the work. All authors discussed the results and reviewed the manuscript.

## Additional information

**Competing financial interests:** The authors declare no competing financial interests.

