## [Peer review file · Nature Communications]

Reviewers' Comments:

Reviewer #1 (Remarks to the Author)

The paper "Bright high-order harmonic generation with controllable polarization from a relativistic plasma mirror" describes a numerical investigation into the generation of circular and elliptical high order harmonics from relativistic laser plasma interactions. This has importance for the control of the polarization of the harmonics for a variety of applications. 2D and 1D simulations are performed for a variety of parameters.

Although this is interesting work I am not sure why it has been submitted to Nature Communications. I do not believe that it meets the standards of importance required for publication in this journal. There have already been experiments examining the generation of relativistic harmonics from circularly polarized light; Easter et al. NJP (2013) and Yeung et al. PRL (2015) - the latter paper has not been cited in this manuscript. The 2013 paper also included 3D PIC simulations of harmonics generated by intense laser pulses in this regime.

These experimental papers suggest that the efficiency of harmonic generation from circular polarized light is low.

I suggest that the authors re-consider whether their work agrees with the previous experimental work, add some additional references and re-write the paper for submission to Scientific Reports, PRL or NJP.

Reviewer #2 (Remarks to the Author)

This work describes a novel method for the generation of circularly and elliptically polarised X-UV harmonics from relativistic laser-solid interactions. The idea is to irradiate a solid target with a circularly polarised relativistic intense laser pulse at oblique angles of incidence. With normal incidence irradiation of solid targets with circularly polarised light, it is not possible to generate reflected harmonics, due to the absence of the oscillating component of the ponderomotive force. However, as the authors show here, when these pulses irradiate the target at oblique angles of incidence, then it is possible to decompose the incident laser pulse into their s- and p- polarised elements, which do contain these oscillating components. The authors show via sophisticated computer simulations that, with suitable selection of the incident polarisation, it is possible to generate harmonics with similar polarisation properties.

The manuscript is very interesting contribution to physics. However, I am not convinced that the authors can claim that this mechanism is related to the $-8/3$ scaling of harmonic intensity that they quote. That scaling arises from the theory of relativistic spikes and has been shown to be valid in the strongly relativistic regime. It is related to the zero-vector potential mechanism for absorption of the laser energy (T. Baeva et al., Phys. Plasmas 18, 056702, 2011). The ZVP mechanism relies upon the zeroes in the vector potential travelling through the relativistic skin layer and results in the violent reconstruction of the skin layer that releases the electrons (and X-rays) on the attosecond timescale. Baeva et al. showed that the use of elliptically polarised incident laser light results in the disappearance of harmonics, as the zeroes in the vector potential no longer go through zero, even for cases of oblique angles of incidence.

It appears that the regime described in this manuscript, where arbitrary polarised harmonic generation that the authors have identified, is only relevant to the weakly relativistic regime related to the relativistic oscillating mirror model that they have explored computationally. The regime of validity for this mechanism, as well as the coherent lighthouse emission regime described by J. Wheeler et al., Nature Photonics 6, 829-833 (2012) and the zero vector potential mechanism must be addressed in the revised manuscript.

Overall, the work is interesting, novel and the methodology appears to be sound, provided that the range of validity issues described above are addressed. I think that this work will spur a lot of interest in the academic community to verify the results, particularly as harmonic filtering will generate attosecond pulse trains. It could become a highly cited paper as a result.

RESPONSES TO REFEREES' COMMENTS

We would like to thank the editor and anonymous referees for the helpful suggestions and critical and constructive comments on an earlier version of the paper. The paper has been revised in line with all of these suggestions and comments.

Reviewer #1:

Specific Comments:

1. *“There have already been experiments examining the generation of relativistic harmonics from circularly polarized light; Easter et al. NJP (2013) and Yeung et al. PRL (2015) - the latter paper has not been cited in this manuscript.”*

We thank the reviewer for this helpful comment. It should be noted that both of the two papers are *not* about the generation of circular and elliptical high-order harmonics, *nor* have they measured or discussed the polarization state of the harmonics.

Instead, the paper by Easter et al. (NJP 2013) only measured the angular intensity distribution of the harmonics by using circular laser pulses. While the paper by Yeung et al. (PRL 2015) only measured the harmonic intensity as a function of the laser ellipticity.

In this sense, these two works have not touched upon the key points of our manuscript. Therefore, they do not affect the extent of novelty of our findings.

List of actions:

To establish the advance of our work over previous works more clearly, we have elaborated the following discussions in the introduction section of the revised manuscript (see the second paragraph in page 2):

“For EP laser pulse at oblique incidence, two experimental groups observed HHG at relatively small incidence angles (Easter et al. NJP 2013; Yeung et al. PRL 2015). However, only the harmonic intensities were measured, with no information about the harmonic polarization states.”

2. *“The 2013 paper also included 3D PIC simulations of harmonics generated by intense laser pulses in this regime.”*

We thank the reviewer for spotting this. Again, this simulation result has *not* demonstrated the generation of circular and elliptical high-order harmonics. Instead, it is about the angular intensity distribution of the harmonics using circular laser pulses.

Moreover, the presented 3D simulation results are only of the 3rd harmonic (wavelength ~ 200 nm, out of XUV range), due to the computational constraints. On the one hand, such low-order harmonic is not of much interest for the applications we discussed. On the other hand, efficient harmonic generation of much higher orders cannot be simply deduced just from the results of the 3rd harmonic.

3. “These experimental papers suggest that the efficiency of harmonic generation from circular polarized light is low.”

We thank the reviewer for this helpful comment. The efficiency of harmonic generation from circularly polarized (CP) light is low *only* when it is at normal incidence or small-angle oblique incidence. However, with increasing the incidence angle, the efficiency increases rapidly and can be comparable with using linearly polarized (LP) light.

The experimental results by Yeung et al. (PRL 2015) showed, at 22.5° incidence, the harmonic (13th-28th) efficiency using CP laser is at least *two order* of magnitude lower than that using LP laser.

While the experimental results by Easter et al. (NJP 2013) showed, at 35° incidence, the harmonic (13th-19th) efficiency using CP laser is just *a factor of 3* lower than that using LP laser.

To verify the efficiency dependence on the incidence angle, we carried out a series of 1D PIC simulations. The results are shown in Fig. R1.

Figure R1. The simulated influence of incidence angle on the efficiency of harmonic (13th-30th) generation from circularly and linearly polarized laser pulses. In these simulations, the laser amplitude is $a_0=5$ for CP laser and $a_0=7.07$ for LP laser to keep the intensity and pulse energy the same. The plasma density is $n_e=200n_c$ and scale length is $L_s=0.1$.

We see that the simulation results at incidence angles of 22.5° and 35° are in excellent agreement with the above experimental results. Besides, the simulation results predict the harmonic efficiency with CP laser increases with increasing the incidence angle, which reaches the same value as using LP laser at 45° incidence.

Further increasing the incidence angle, the two efficiencies tend to stay at the same level. Figure 5(b) in the manuscript shows the E_y components of the HHG spectra by using CP and LP lasers at $\theta = 55^\circ$ incidence. It is also seen the HHG efficiency with CP laser is comparable to that with LP laser. These results show the potential of achieving efficient helical HHG with our scheme. Moreover, as mentioned, intense HHG can be generated even with a low efficiency, since the applicable laser intensity is high.

List of actions:

In the revised manuscript, we have incorporated the above discussions on HHG efficiency to demonstrate that our scheme can achieve high efficiency. Please see the subsection “HHG efficiency” in page 6.

General Comment:

1. *“This has importance for the control of the polarization of the harmonics for a variety of applications. 2D and 1D simulations are performed for a variety of parameters. Although this is interesting work I am not sure why it has been submitted to Nature Communications. I do not believe that it meets the standards of importance required for publication in this journal. I suggest that the authors re-consider whether their work agrees with the previous experimental work, add some additional references and re-write the paper for submission to Scientific Reports, PRL or NJP.”*

We thank the reviewer for this critical comment. We note that XUV sources with controllable polarization are indeed highly demanded among a large variety of scientific communities yet challenging to produce. One example is the specially designed FEL facility FERMI to achieve this. Although limited and available quite recently, many users from different areas have already taken advantage of this unique capability to perform insightful research.

It can be conceived that a much larger avenue of research will get benefit, if such light sources have a wider accessibility. Thus to achieve such source with smaller-scale and lower-cost is of great importance.

Researchers working on HHG from laser gas interactions have made much effort towards such tabletop sources, with partial success very recently. Over the past year, several papers are published on Nature’s sister journals on this topic, e.g.,

Kfir, O. *et al.* Generation of bright phase-matched circularly-polarized extreme ultraviolet high harmonics. *Nature Photon.* 9, 99-105 (2015).

Ferre, A. *et al.* A table-top ultrashort light source in the extreme ultraviolet for circular dichroism experiments. *Nature Photon.* 9, 93-98 (2015).

Hickstein, D. D. *et al.* Non-collinear generation of angularly isolated circularly polarized high harmonics. *Nature Photon.* 9, 743-750 (2015).
Cireasa, R. *et al.* Probing molecular chirality on a sub-femtosecond timescale. *Nature Phys.* 11, 654-658 (2015).
Lambert, G. *et al.* Towards enabling femtosecond helicity-dependent spectroscopy with high-harmonic sources. *Nature Commun.* 6, 6167 (2015).
Fleischer, A. *et al.* O. Spin angular momentum and tunable polarization in high-harmonic generation. *Nature Photon.* 8, 543-549 (2014).

These papers show the great importance and large interest of such work. Though inspiring, the photon yields from these sources remain low. For photon-demanding applications like single-shot measurements, photon-rich CP/EP XUV pulses are crucial.

To solve this riddle, our new laser-plasma scheme provides a very promising and straightforward route to achieve bright source with controllable polarization. It has the potential and impact to fill the gap between large-scale facilities and HHG from gas. It is thus of broad interest for a large variety of research communities, e.g., chemical, biological, medical, condensed matter and material sciences.

Besides, the physics involved is also of great interest for specialists within the laser plasma field. As was presumed, CP laser could not generate HHG from relativistically oscillating mirror (ROM). Both reviewers also had such concerns. However, we show for the first time this is not the case if the laser-plasma parameters are suitable. In addition to reproduce the previous experimental results, our simulation results also predict the harmonics can be efficiently generated and the mechanism is within the ROM regime. It is hoped that the present work will stimulate future experiments to verify the interesting results.

With this significance and broad interest, our work represents important advances towards reaching polarization controllable XUV with high brilliance suitable for a large avenue of research. Thus we believe it meets the standards of importance for publication in Nature Communications. We truly hope this paper will spur a lot of interest among the communities, and eventually benefit a wide range of research in turn.

We thank the reviewer again and hope all of the above changes would improve the presentation of the paper and give the reader a clearer picture.

Reviewer #2:

General Comment:

1. *“Overall, the work is interesting, novel and the methodology appears to be sound, provided that the range of validity issue described above are addressed. I think that this work will spur a lot of interest in the academic community to verify the results, particularly as harmonic filtering will generate attosecond pulse trains. It could become a highly cited paper as a result.”*

We thank the reviewer for this favorable comment. The paper has been revised in line with all of the comments given below.

Specific Comments:

1. *“The manuscript is very interesting contribution to physics. However, I am not convinced that the authors can claim that this mechanism is related to the -8/3 scaling of harmonic intensity that they quote. That scaling arises from the theory of relativistic spikes and has been shown to be valid in the strongly relativistic regime. It is related to the zero-vector potential mechanism for absorption of the laser energy (T. Baeva et al., Phys. Plasmas 18, 056702, 2011). The ZVP mechanism relies upon the zeroes in the vector potential travelling through the relativistic skin layer and results in the violent reconstruction of the skin layer that releases the electrons (and X-rays) on the attosecond timescale. Baeva et al. showed that the use of elliptically polarised incident laser light results in the disappearance of harmonics, as the zeroes in the vector potential no longer go through zero, even for cases of oblique angles of incidence.*

It appears that the regime described in this manuscript, where arbitrary polarised harmonic generation that the authors have identified, is only relevant to the weakly relativistic regime related to the relativistic oscillating mirror model that they have explored computationally. The regime of validity for this mechanism, as well as the coherent lighthouse emission regime described by J. Wheeler et al., Nature Photonics 6, 829-833 (2012) and the zero vector potential mechanism must be addressed in the revised manuscript.”

We thank the reviewer for this thoughtful comment. As we showed in the methods section, the transverse momentum \mathbf{P}_\perp of plasma electrons in the case of oblique incidence by using CP laser can be expressed as

$$\mathbf{p}_\perp = (-m_e c \tan \theta + eA_0 / c \sin \omega_0 t) \mathbf{e}_y - eA_0 / c \cos \omega_0 t \mathbf{e}_z$$

At normal incidence, $\theta = 0$. Then we have

$$\mathbf{p}_\perp = eA_0 / c (\sin(\omega_0 t) \hat{e}_y - \cos(\omega_0 t) \hat{e}_z)$$

In this case, the zeros in the transverse momenta no longer go through zero. Therefore, according to the Υ -spikes model, the harmonic generation disappears at normal incidence.

However, under oblique incidence, $\theta \neq 0$. And thus \mathbf{P}_\perp can go through zero even for the case of oblique incidence using elliptical laser.

To demonstrate this, we plot the electron phase space (p_x, x) and the transverse momenta \mathbf{P}_\perp distribution at two different times from the simulation results with CP laser, similar with that using LP laser presented in Fig.2 in the paper by Baeva *et al.* (Phys. Plasmas 18, 056702, 2011). See Fig. R2.

Figure R2. Electron phase space (p_x, x) and transverse momenta (p_y and p_z) distribution for the case of CP laser at oblique incidence at two moments of (top) $t=28.64T_0$ and (bottom) $t=30.40T_0$. In these simulations, the CP laser amplitude is $a_0=5$, the plasma density is $n_e=200n_c$, scale length is $L_s=0.1$, and the incidence angle is 55° . The momenta are normalized by $m_e c$.

It can be clearly seen that time moments do exist when the both transverse momenta of plasma surface electrons become zero simultaneously. According to the Υ -spikes model, it is at these moments that the harmonics are efficiently emitted.

Besides, both the 2D and 1D simulation results have shown that the theoretically predicted $-8/3$ scaling law does fit the harmonic spectra well. Please see Fig. 1 and Fig. 5 in the revised manuscript.

In addition, the parametric studies also show our scheme works well for a broad range of laser amplitudes (a_0 from 5 to 30) at oblique incidence.

These results suggest the mechanism here is in accordance with the relativistically oscillating model (ROM), more specifically, the Υ -spikes model, and thus the simulations are relevant to the strongly relativistic regime with $a_0^2 \gg 1$ and to relativistically overdense plasmas with $S \gg 1$ (S is the dimensionless similarity parameter).

We also note that in the paper by Baeva *et al.* (Phys. Plasmas 18, 056702, 2011), the results for the case of using elliptical incident laser are at normal incidence (see section III D and Fig. 8). The results at oblique incidence are for LP laser, when discuss the fast electrons traveling in the direction of laser propagation (see section III E and Fig. 9).

Regarding the attosecond lighthouse effect, it is actually not a HHG generation mechanism, but a concept to generate angularly separated attosecond pulses so that in a specific direction a single isolated attosecond pulse can be obtained. The idea is to use an incident laser with a rotating wavefront. Therefore, it can in principle be applied to any HHG mechanism. However, due to some difficulties, it can only be achieved with few-cycle (e.g. ≤ 5 fs) laser pulses at present, which have relatively low intensities.

As for the paper by J. Wheeler *et al.* (Nature Photonics 6, 829-833 (2012)), due to the low laser intensity used, the dominant HHG mechanism works in the coherent wake emission (CWE) regime. This mechanism is known to be efficient for weakly relativistic interactions $a \leq 1$ and only limited to harmonic frequencies less than the plasma frequency. As such, these are not the cases for our scheme.

List of actions:

In the revised manuscript, we have elaborated the above discussions to illustrate the regime of validity and the associated emission mechanism more clearly. Please see the subsection “regime of validity” in page 6.

We also added the paper by J. Wheeler *et al.* (Nature Photonics 6, 829-833 (2012)) as a reference for the coherent wake emission (CWE) mechanism. Besides, we have addressed their regimes of validity respectively: CWE, dominant in weakly relativistic regime; ROM, dominant in strongly relativistic regime. Please see paragraph one in page 2.

We thank the reviewer again and hope all of the above changes improve the overall presentation of the paper and give the reader a clearer picture.

REVIEWERS' COMMENTS:

Reviewer #1 (Remarks to the Author):

The changes in the paper and the responses to my previous comments are sufficient for me to now recommend acceptance of this paper for publication in Nature Communications.

Reviewer #2 (Remarks to the Author):

I have read through the response to my initial report on this manuscript. I am very happy with the changes made and recommend publication of the manuscript as it now stands.

RESPONSES TO REFEREES' COMMENTS

Reviewer #1:

“The changes in the paper and the responses to my previous comments are sufficient for me to now recommend acceptance of this paper for publication in Nature Communications.”

We would like to thank the anonymous referee for this favorable remark. We appreciate his/her useful suggestions and critical and constructive comments on the earlier versions of the paper, which are of great help in improving the manuscript.

Reviewer #2:

“I have read through the response to my initial report on this manuscript. I am very happy with the changes made and recommend publication of the manuscript as it now stands.”

We would like to thank the anonymous referee for this favorable remark. We appreciate his/her useful suggestions and critical and constructive comments on the earlier versions of the paper, which are of great help in improving the manuscript.